# A Hybrid Binary Dragonfly Algorithm with an Adaptive Directed Differential Operator for Feature Selection

Yilin Chen [1,2,3], Bo Gao [1], Tao Lu [1,2], Hui Li [1,2], Yiqi Wu [4], Dejun Zhang [4] and Xiangyun Liao [5,*]

1 School of Computer Science and Engineering, Wuhan Institute of Technology, Wuhan 430073, China; yilinchen@wit.edu.cn (Y.C.); bogao@stu.wit.edu.cn (B.G.); lutxyl@gmail.com (T.L.); lihui00317@163.com (H.L.)
2 Hubei Key Laboratory of Intelligent Robot (Wuhan Institute of Technology), Wuhan 430073, China
3 Hubei Engineering Research Center of Intelligent Production Line Equipment (Wuhan Institute of Technology), Wuhan 430073, China
4 School of Computer Science, China University of Geosciences, Wuhan 430073, China; wuyq@cug.edu.cn (Y.W.); zhangdejun@cug.edu.cn (D.Z.)
5 Shenzhen Institute of Advanced Technology, Chinese Academy of Sciences, Shenzhen 518052, China
* Correspondence: xy.liao@siat.ac.cn

**Abstract:** Feature selection is a typical multiobjective problem including two conflicting objectives. In classification, feature selection aims to improve or maintain classification accuracy while reducing the number of selected features. In practical applications, feature selection is one of the most important tasks in remote sensing image classification. In recent years, many metaheuristic algorithms have attempted to explore feature selection, such as the dragonfly algorithm (DA). Dragonfly algorithms have a powerful search capability that achieves good results, but there are still some shortcomings, specifically that the algorithm's ability to explore will be weakened in the late phase, the diversity of the populations is not sufficient, and the convergence speed is slow. To overcome these shortcomings, we propose an improved dragonfly algorithm combined with a directed differential operator, called BDA-DDO. First, to enhance the exploration capability of DA in the later stages, we present an adaptive step-updating mechanism where the dragonfly step size decreases with iteration. Second, to speed up the convergence of the DA algorithm, we designed a new differential operator. We constructed a directed differential operator that can provide a promising direction for the search, then sped up the convergence. Third, we also designed an adaptive paradigm to update the directed differential operator to improve the diversity of the populations. The proposed method was tested on 14 mainstream public UCI datasets. The experimental results were compared with seven representative feature selection methods, including the DA variant algorithms, and the results show that the proposed algorithm outperformed the other representative and state-of-the-art DA variant algorithms in terms of both convergence speed and solution quality.

**Keywords:** feature selection; binary dragonfly algorithm; differential evolution algorithm; multiobjective optimization; classification

## 1. Introduction

With the advances in data mining and machine learning techniques, many research problems now involve the analysis of multiple datasets. These datasets frequently contain numerous irrelevant, noisy, and/or redundant features, which significantly affect the classification performance. Specifically, irrelevant or noisy features can significantly degrade the classification performance due to their misleading information [1]. Feature selection (FS) aims to enhance classification performance, reduce data dimensionality, save storage space, improve computational efficiency, and facilitate data visualization and understanding by selecting a small subset of relevant features. The increase in data and dimensionality leads to an increasingly difficult feature selection [2].

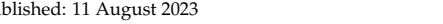

In recent years, feature selection has found widespread application in various research domains, including text classification, remote sensing [3], intrusion detection, gene analysis, and image retrieval [4]. Notably, feature selection plays a crucial role in remote sensing image classification tasks. Remote sensing images often contain a vast amount of pixel data and diverse spectral information [5,6], yet only a subset of features contributes significantly to the classification process. Hence, feature selection techniques are used to identify the most informative and relevant features, reduce the complexity of the classification tasks, and improve the overall accuracy. By effectively selecting pertinent features, the interference of redundant information and noise can be minimized, while emphasizing the spatial, spectral, and textural characteristics of objects in remote sensing images. Consequently, feature selection holds paramount significance in achieving precise land object classification and monitoring in remote sensing applications [7,8]. To this end, the development of efficient feature selection methods becomes imperative, aiming to optimize classification accuracy and computational efficiency.

Feature selection methods are mainly divided into two categories: filters and wrappers [9]. Filter-based methods use the intrinsic information of features, such as correlation, information gain, and consistency [10], to determine features [11]. The wrapper-based approach is used to generate feature subsets through a classifier and a learning model and perform accuracy evaluation by a learning algorithm to determine the feature subsets [12,13]. Although wrapper-based methods can produce better feature subsets, they are computationally expensive, especially when the search space is large and complex. In contrast, filter-based methods are usually less computationally expensive, but have lower classification accuracy and cannot select a good subset of features [14].

Although feature selection is an indispensable process, it is also a complex and challenging combinatorial optimization problem. The challenges of feature selection mainly revolve around three aspects. Firstly, the search space for feature selection exponentially grows with the number of features, as $N$ features can yield $2^N$ feature subsets [15,16]. Therefore, an exhaustive search of all possible subsets is impractical, especially when dealing with numerous features. Secondly, features have complex interactions with each other. Thirdly, feature selection is inherently a multiobjective problem. The two primary objectives of feature selection are to maximize classification performance and to minimize the number of selected features. However, these objectives are often conflicting [17]. A robust and powerful search algorithm forms the foundation for addressing feature selection problems effectively.

In recent years, metaheuristics have been successfully applied to feature selection methods. Metaheuristic algorithms are inspired by natural phenomena, such as particle swarm optimization (PSO) [18–20], genetic algorithms (GA) [21], the differential evolution algorithm (DE) [22,23], the bat algorithm (BA) [24,25], the gray wolf optimization algorithm (GWO) [26–28], the dragonfly algorithm (DA) [29], cuckoo search (CS) [30], the salp swarm algorithm (SSA) [31], Harris hawks optimization (HHO) [32], etc. When using or designing metaheuristic algorithms, there needs to be an effective way to maintain a balance between development and exploration. Metaheuristic algorithms need to search as much of the search space as possible in the early stages and develop the optimal region in the later stages, so a balance between exploration and development is essential.

The dragonfly algorithm (DA) is a recently developed optimization algorithm inspired by the collective behavior of dragonflies in nature. Initially designed for continuous optimization tasks, a binary version called the binary dragonfly algorithm (BDA) was later introduced to address discrete problems [33]. While BDA has shown strong performance on various datasets, it may suffer from limited exploration capabilities, potentially leading to local optimization problems.

In this study, we propose a hybrid method that combines the BDA algorithm with the directed differential operator to improve the performance of BDA. An adaptive step-updating mechanism is also proposed to enhance the exploration capability of the algorithm which improves the search performance. The proposed method has been tested on

14 mainstream datasets and compared with seven representative feature selection methods, including the DA variant algorithms. The main contributions of this study are as follows:

- The algorithm incorporates an adaptive step-updating mechanism to optimize the search process. It adjusts the step size based on the stage of exploration: a longer step size benefits an early search, while a smaller step size is advantageous for later exploration. To achieve this, the dragonfly's step size gradually decreases with each iteration, adapting to the evolving search landscape. This adaptive mechanism enhances the algorithm's ability to balance exploration and exploitation, resulting in improved overall performance.
- Based on the renowned differential evolution algorithm, we propose a directed differential operator. This operator utilizes information from both the best and worst positions of the dragonflies to guide the algorithm towards promising regions, facilitating more effective search and convergence to optimal solutions. Additionally, the size of the directed differential operator influences the balance between targeted exploitation and broad exploration. A larger operator emphasizes guidance, while a smaller one allows for greater exploration of the solution space. To achieve this balance, we design an adaptive updating mechanism to adjust the directed differential operator, playing a crucial role in optimizing exploration and exploitation within the search space.
- We enhance BDA's exploration capability by integrating it with the differential evolution algorithm (DE). During the position-updating phase of the dragonfly algorithm, individual positions are combined with the directed differential operator to guide the search in promising directions, accelerating convergence. To maintain population diversity, we introduce an adaptive method for updating the directed differential operator. Additionally, an adaptive update mechanism for the dragonfly step dynamically adjusts the step size to improve exploration in later stages. This integration significantly improves BDA's performance and exploration capabilities.

The paper's structure is as follows: In Section 2, related works are discussed, focusing on recent feature selection algorithms and the binary dragonfly algorithm (BDA). Section 3 provides a detailed explanation of the proposed hybrid BDA-DDO algorithm, including the three improvement mechanisms and the algorithm's specific process. Section 4 describes the experimental setup and analysis of the obtained results. Finally, Section 5 presents the conclusion and proposes future research directions.

## 2. Related Work

In this section, an extensive review of feature selection algorithms is presented, with a specific emphasis on the binary dragonfly algorithm (BDA). We provide a comprehensive overview of existing approaches to feature selection, highlighting their methodologies and performance. Furthermore, we delve into the intricacies of BDA, elucidating its principles and underlying mechanisms.

### 2.1. Related Work on Feature Selection

Metaheuristic algorithms have gained popularity in the past decade due to their advantages, and they have been widely applied to feature selection and optimization problems. As a result, several feature selection algorithms based on metaheuristics have been proposed. This section presents a comprehensive review of some noteworthy feature selection algorithms that have emerged in recent years [14].

In [34], Xue et al. proposed a feature selection approach that combines multiobjective particle swarm optimization (PSO) with NSGA-II. The algorithm efficiently explores the solution space by integrating the nondominated sorting mechanism into PSO, resulting in a set of nondominated solutions rather than a single one. However, a limitation of the method is the rapid loss of diversity in the later stages, which affects its feature selection performance. The particle-ranking-based PSO method [35] divides the objective space into several subregions using uniform and nonuniform partitioning techniques. It calculates

particle rank and feature rank to update particle velocity and position in each generation. This approach accelerates optimization and guides the particles to better solutions. Luo et al. [36] proposed a memetic algorithm based on particle swarm optimization to tackle high-dimensional feature selection. The algorithm includes information-entropy-based initialization and adaptive local search techniques to enhance search efficiency. Additionally, a novel velocity-updating mechanism is introduced to consider solution convergence and diversity. The algorithm demonstrates strong performance in improving feature subset quality and size. The algorithm's performance on low-dimensional datasets may be limited, necessitating further enhancement of its robustness.

Xue et al. [37] introduced a hybrid algorithm that combines PSO and DE for feature selection. DE is used to generate promising individuals to guide the search process. This approach maintains diversity and directs particles to promising regions, enabling effective escape from local optima. The algorithm performs well on small-scale datasets, but its performance on large-scale datasets remains unverified. Nelson et al. [22] introduced a hybrid DE algorithm that combines binary differential evolution (BDE) for obtaining feature subsets and incorporates a local search method to minimize classification error. The algorithm's essence lies in using a classical feature selection algorithm as the local search strategy to enhance the solutions derived from DE. In this way, the hybrid method aims to strike a balance between exploration and exploitation in achieving improved solutions. In their work [38], Wang et al. proposed SaWDE, a novel weighted differential evolution algorithm that addresses large-scale feature selection problems. SaWDE employs a multipopulation mechanism for enhanced diversity and introduces an adaptive strategy to capture diverse features from historical dataset information. Additionally, a weighted model is used to identify important features, leading to effective feature selection solutions. The algorithm performs remarkably well in reducing feature dimensions and demonstrates strong performance on large-scale datasets. However, further research is needed to fine-tune the number of subpopulations and relevant parameters in the algorithm. Xue et al. proposed a multiobjective differential evolution approach in [39] to search for multiple optimal feature subsets. They used a method that considers feature correlations for initialization to provide a strong starting point. The population was divided into several subarchives using a clustering approach, and within each subarchive, a sophisticated crowding distance was utilized to ensure diversity by considering both the search and objective spaces. Nondominated solutions from all subarchives were stored in another archive to guide the evolutionary feature selection process. The proposed algorithm achieved improved feature subsets, but the training process may take longer due to the increased computational time.

The dragonfly algorithm (DA) is a recent metaheuristic algorithm introduced by Mirjalili et al. In [33], they proposed a wrapper feature selection algorithm based on the binary dragonfly algorithm (BDA). This algorithm uses a classical transfer function to convert the continuous search space into a discrete space, which effectively solves feature selection problems and outperforms particle swarm optimization (PSO) and genetic algorithms (GA). In their subsequent work [40], Mirjalili et al. introduced three different update mechanisms to balance exploration and exploitation in the dragonfly algorithm. The updated algorithm showed improved performance compared to the binary dragonfly algorithm. The reduced exploration capability in the later stages of the algorithm is still a limitation, leading to the possibility of encountering local optimizations. Too et al. [41] proposed the hyper-learning binary dragonfly algorithm (HLBDA) to mitigate the binary dragonfly algorithm's susceptibility to local optima. The algorithm employs a hyper-learning strategy to improve search behavior and avoid getting trapped in local optimization. HLBDA was applied to the COVID-19 dataset, and the results show its superior performance in enhancing prediction accuracy. Hamouda et al. [42] proposed a hybrid feature selection algorithm that combines the dragonfly algorithm (DA) with simulated annealing to address the issue of local optimization in DA. By integrating simulated annealing, the algorithm aims to improve its ability to select the optimal feature subset for classification tasks and mitigate the problem of local optima. The approach shows promising results in enhancing

classification accuracy. Duan et al. [43] proposed hybrid DA-DE, a novel algorithm that combines the dragonfly algorithm (DA) and differential evolution (DE) to address global optimization problems. The algorithm introduces a mutation operator from DA and adapts the scaling factor in an individual-dependent and self-adaptive manner. By integrating DE's development capability and DA's exploration ability, the algorithm achieves optimal global solutions effectively, especially on high-dimensional functions.

In response to the limitations of the dragonfly algorithm (DA) and based on previous research progress, we propose a hybrid approach by combining the binary dragonfly algorithm with the directed differential operator. Our goal is to enhance the algorithm's search capability and improve its performance in feature selection problems.

### 2.2. Binary Dragonfly Algorithm

The Dragonfly algorithm (DA) is a swarm intelligence optimization algorithm proposed by Seyedali Mirjalili [29] in 2015, inspired by the swarm intelligent behavior of dragonflies. It was observed that dragonfly swarming behavior comprises two main patterns: hunting and migration [29,42]. The hunting mechanism involves dragonflies swarming in search of food sources, while the migration mechanism involves large groups of dragonflies migrating over long distances in a specific direction. In 2016, Mirjalili introduced the binary dragonfly algorithm (BDA), which is a discrete version based on the original DA and adapted for binary optimization problems.

The dragonfly algorithm employs five primary behaviors, which is essential for updating the dragonfly positions. Each of these behaviors is described as follows:

- *Separation*, in order to avoid static collisions between individuals and other nearby individuals. The mathematical model of separation behavior is represented by Equation (1).

$$S_i = -\sum_{j=1}^{N} X - X_j \tag{1}$$

  In the presented model, $X$ represents the current search agent, and $X_j$ denotes its $j$-th neighbor. The parameter $N$ indicates the total number of neighbors.

- *Alignment*, which represents the speed match between the current individual and other surrounding individuals. The mathematical model can be represented by Equation (2).

$$A_i = \frac{\sum_{j=1}^{N} V_j}{N} \tag{2}$$

  where $V_j$ denotes the velocity of the $j$-th neighbor.

- *Cohesion*, which implies that the current individual tends to move closer to the mass center of the surrounding group. Mathematically, this cohesion behavior is expressed by Equation (3).

$$C_i = \frac{\sum_{j=1}^{N} x_j}{N} - X \tag{3}$$

- *Attraction*, which entails the attraction of a food source to an individual and its movement towards the food source. Mathematically, it is expressed by Equation (4).

$$F_i = F_{location} - X \tag{4}$$

  where $F_{location}$ represents the location of the food source.

- *Distraction*, which means the individual moves outward away from the enemy's position. The behavior of the $i$-th individual principle enemy is represented by Equation (5).

$$E_i = E_{location} + X \tag{5}$$

  where $E_{location}$ represents the location of the current enemy.

The dragonfly algorithm follows a framework similar to that of particle swarm optimization (PSO). The position update process primarily involves two vectors: the step vector ($\Delta X$) and the position vector ($X$). The step size vector serves a role similar to the speed vector in PSO, determining both the movement direction and step size of the dragonfly. The step vector is defined as follows:

$$\Delta X_{t+1} = (sS_i + aA_i + cC_i + fF_i + eE_i) + \omega \Delta X_t \tag{6}$$

where $s$ is the separation weight, $a$ is the alignment weight, $c$ is the cohesion weight, $f$ is the food weight, $e$ is the enemy weight, $\omega$ is the inertia weight, and $t$ is the current number of iterations. Equation (7) demonstrates adaptively adjusting the value of $\omega$ to balance exploration and exploitation. Hammouri et al. [40] proposed three adaptive methods to adjust these parameters, aiming to enhance BDA's performance. Further explanation of these adaptive methods will be provided below.

$$\omega = 0.9 - Iter * \frac{(0.9 - 0.4)}{Max\_iter} \tag{7}$$

The position of an individual is updated as in Equation (8):

$$X_{t+1} = X_t + \Delta X_{t+1} \tag{8}$$

In LBDA, a linear model is utilized to update parameter values within specific ranges, as shown in Equation (9). The values of $s$, $a$, and $c$ are linearly updated in the range of 0 to 0.2. Meanwhile, the value of $e$ is updated within the range of 0.1 to 0, and the value of $f$ is updated in the range of 0 to 2.

$$
\begin{aligned}
s &= 0.2 - (0.2 \times iter/max\_iter) \\
e &= 0.1 - (0.1 \times iter/max\_iter) \\
a &= 0.0 + (0.2 \times iter/max\_iter) \\
c &= 0.0 + (0.2 \times iter/max\_iter) \\
f &= 0.0 + (2.0 \times iter/max\_iter)
\end{aligned}
\tag{9}
$$

In QBDA, a quadratic model is utilized to update the value of each parameter in a nonlinear manner. The parameter value range remains consistent with that of LBDA.

$$
\begin{aligned}
s &= (0.2 - (0.2 \times iter/max\_iter))^2 \\
e &= (0.1 - (0.1 \times iter/max\_iter))^2 \\
a &= (0.0 + (0.2 \times iter/max\_iter))^2 \\
c &= (0.0 + (0.2 \times iter/max\_iter))^2 \\
f &= (0.0 + (2.0 \times iter/max\_iter))^2
\end{aligned}
\tag{10}
$$

In SBDA, a sinusoidal model is used to update the values of the coefficients in a nonlinear manner. The parameter value range remains the same as that of LBDA.

$$
\begin{aligned}
s &= 0.10 + (0.10 \times |\cos(iter/max\_iter \times 4 \times \pi - \beta)|) \\
e &= 0.05 + (0.05 \times |\cos(iter/max\_iter \times 4 \times \pi - \beta)|) \\
a &= 0.10 - (0.05 \times |\cos(iter/max\_iter \times 4 \times \pi - \beta)|) \\
c &= 0.10 - (0.05 \times |\cos(iter/max\_iter \times 4 \times \pi - \beta)|) \\
f &= 2.00 - (1.00 \times |\cos(iter/max\_iter \times 4 \times \pi - \beta)|)
\end{aligned}
\tag{11}
$$

Feature selection poses distinct challenges in binary and continuous search spaces due to its discrete nature. In the continuous search space, the position update is achieved by

adding the updated step vector to the position vector. However, in the discrete search space, where the position vector can only take binary values (0 or 1), a transfer function is essential to convert the continuous space into the discrete space. BDA adopts a *V*-shaped transfer function, represented by the following equation, where $X_i^d(t)$ represents the d-dimensional position of the *i*-th dragonfly, and $\Delta X$ represents the step vector.

$$X_i^d(t+1) = \begin{cases} 1 - X_i^d(t) & rand < TF(\Delta X_i^d(t+1)) \\ X_i^d(t) & rand \geq TF(\Delta X_i^d(t+1)) \end{cases} \tag{12}$$

$$TF(\Delta X) = \left| \frac{\Delta X}{\sqrt{\Delta X^2 + 1}} \right| \tag{13}$$

## 3. The Proposed Feature Selection Method

In this section, we present the binary dragonfly algorithm with a hybrid directed differential operator (BDA-DDO) and its application to feature selection. While BDA exhibits powerful search capability, it faces challenges such as limited late exploration, slow convergence, and a lack of population diversity. To effectively address these issues, we propose three improvement mechanisms. In BDA-DDO, we introduce a directed differential operator that combines BDA-generated individuals with the differential operator, resulting in faster convergence. Additionally, we propose an adaptive approach to update this operator, thereby increasing population diversity. Furthermore, we design an adaptive step-updating mechanism to promote exploration in later stages. We provide a detailed explanation of these improvements and describe the fitness function used to evaluate the quality of the solution.

### 3.1. Directed Differential Operator

Differential evolution (DE), introduced by Storn and Price [44] in 1997, has gained significant attention from researchers due to its simple structure, fast convergence speed, and ease of implementation. In order to enhance the convergence speed of BDA, we have devised a new differential operator based on the DE algorithm. This novel differential operator combines the optimal and worst solutions of the dragonfly, forming a directed differential operator. By doing so, it provides a promising direction in the search space, guiding individuals towards better solutions and facilitating faster convergence to the optimal region. Below, we provide a step-by-step description of the implementation of the directed differential operator.

- Mutation: In the proposed algorithm, mutation is applied to the *i*th dragonfly individual to generate a mutation vector. Storn and Price [45] introduced several mutation operations, with "DE/rand/1" being the most typical one:

$$V_i = X_i + F \times (X_{Food} - X_{Enemy}) \tag{14}$$

  Among them, $X_i$ represents the position of the individual after updating through BDA, $X_{Food}$ represents the location of the dragonfly's food (the best location in history), and $X_{Enemy}$ represents the location of dragonfly enemies. The scaling factor, denoted by $F$, controls the amplification of the difference vector and significantly influences the convergence speed. We will provide a detailed explanation of $F$ later in this section.

- Crossover: After mutation, a crossover operation is performed by randomly selecting either the mutant individual $V_{i,j}$ or the original individual $X_{i,j}$ to generate the experimental individual. Among the three classical crossover operators [46]—binomial crossover, exponential crossover, and rotation-invariant arithmetic crossover—we are using the binomial crossover operator, as shown in the following equation:

$$U_{i,j} = \begin{cases} V_{i,j} & rand < CR \quad or \quad randi(1,d) = j \\ X_{i,j} & rand > CR \quad or \quad randi(1,d) \neq j \end{cases} \tag{15}$$

where *rand* is a random variable between [0, 1] and *jrand* is an integer randomly selected from [1, *D*], ensuring that at least one of $U_{i,j}$ comes from $V_{i,j}$. *CR* is the crossover probability, selected from [0, 1], which controls the population diversity. After the crossover operation, the resulting $U_{i,j}$ vector is transformed into a discrete space vector using the transfer function *T1*. Then, a selection operation is carried out to determine whether it can survive to the next generation. The transfer function *T1* is defined as follows:

$$T1 = \frac{1}{1 + e^{-(U_i^d)}} \tag{16}$$

$$U_{i,j} = \begin{cases} U_{i,j} = 0 & rand < T1 \\ U_{i,j} = 1 & rand > T1 \end{cases} \tag{17}$$

- Selection: After the mutation and crossover operations, selection is performed to determine the survival of individuals $U_i$ and $X_i$ in the next generation.

$$X_i = \begin{cases} U_i & f(U_i) \leq f(X_i) \\ X_i & others \end{cases} \tag{18}$$

where $f(U_i)$ and $f(X_i)$ represent the fitness functions corresponding to $U_i$ and $X_i$, respectively.

### 3.2. Time-Varying Differential Vector

To address the issue of population diversity in the late stage of the BDA algorithm, we propose a time-varying differential vector approach. The value of the differential vector decreases from its initial value as iterations progress. The directed differential operator introduced earlier provides directional information. In the early stage, a larger differential vector is used to provide more valuable information for individual search. In contrast, a smaller differential vector is applied in the late stage to improve population diversity in the optimal region, there by improving the overall search performance. The time-varying differential vector is designed as shown in Equation (19), where *F* ranges from [1, 0.5] and gradually decreases with iterations.

$$F = \frac{0.5}{1 + (-0.5) \times e^{-0.5 \times iter}} \tag{19}$$

### 3.3. Adaptive Step-Updating Mechanism

We propose an adaptive step-updating mechanism in the BDA algorithm to address the fixed step size issue. A fixed step size can cause oscillations if too large, or slow convergence if too small, leading to local optima. To overcome this, our adaptive approach dynamically adjusts the dragonfly step size during iterations. Equations (19) and (20) are used for this purpose. The BDA-DDO algorithm, which integrates the adaptive step updating mechanism and the directed differential operator, is given in Algorithm 1. This comprehensive approach aims to enhance exploration and improve population diversity in the late stages of optimization.

$$\Delta X = F \times \Delta X \tag{20}$$

---

**Algorithm 1** Pseudocode of the BDA-DDO

---

**Input:** The number of populations $N$, the maximum number of iterations
**Output:** The best solution

1: Initialize population position $X_i (i = 1, 2, \ldots, N)$
2: Initialize the step vectors, $\Delta X_i (i = 1, 2, \ldots, N)$
3: **while** Maximum number of iterations not reached **do**
4:       Calculate the fitness value of all dragonfly individuals
5:       Update the $F_{location}$, and the $E_{location}$
6:       Update $s, a, c, f, e,$ and $\omega$ (Using Equations (9) or (10) or (11), Using Equation (7))
7:       **for** $i = 1$ to Number of dragonflies, $N$ **do**
8:             Calculate the values of $S, A, C, F, E$ (Using Equations (1)–(5))
9:             Update step vector using Equation (6)
10:            Update step vector using Equation (20)
11:            Update the position of dragonfly (*i*-th) using Equations (12) and (13)
12:       Individual position mutations using Equations (14) and (19)
13:       Crossover operation using Equations (15)–(17)
14:       Select operation using Equation (18)
15: **return** the best solution

---

### 3.4. Fitness Evaluation

Feature selection is a multiobjective problem, where maximum classification accuracy and the minimum subset of features are the goals that need to be achieved. We balance these two objectives with a fitness function, by setting weighting factors. As shown in the following equation [33,40–42]:

$$fitness = \alpha \times ERR + \beta \times \frac{R}{N} \tag{21}$$

where $ERR$ is the classification error rate obtained using the KNN classifier, $R$ is the number of subsets of features selected by the search agent, and $N$ is the total number of features in the datasets. $\alpha$ and $\beta$ are weight factors for balancing classification accuracy and feature subsets. The value range of $\alpha$ is between [0, 1] and the value of $\beta$ is (1-$\alpha$). To maximize the classification accuracy, we set $\alpha$ to 0.99 and $\beta$ to 0.01 in our experiments. We use the parameter settings described in [40] and compare our results with those obtained using the BDA method.

## 4. Experiments and Results Analysis

In this section, we evaluate the performance of three proposed BDA-DDO algorithms, namely LBDA-DDO, QBDA-DDO, and SBDA-DDO, which are the improved versions based on BDA. Section 4.1 presents the details of the mainstream datasets. The parameter information of related algorithms is introduced in Section 4.2. Section 4.3 introduces the performance comparison analysis of the proposed algorithm and LBDA. Section 4.4 presents a comparison between the proposed algorithm and QBDA. Section 4.5 introduces the SBDA-DDO algorithm for comparison with SBDA. Section 4.6 presents a comparison of BDA-DDO with other binary versions of metaheuristic-based feature selection algorithms.

### 4.1. Datasets

In this section, the performance of the proposed algorithm was evaluated on 14 popular datasets collected from the UCI repository [47]. Table 1 provides detailed information about each of the datasets, which are commonly used by researchers to study feature selection methods. From Table 1, it can be observed that the datasets vary in terms of the number of instances, features, and dimensions. This demonstrates that the proposed algorithm's performance has been tested on datasets with different structures. Each dataset in Table 1 was randomly divided into two sets: 80% for training and 20% for testing. To account for the algorithm's randomness, we conducted 30 independent runs for each algorithm.

**Table 1.** Details of datasets.

| DataSet | No. of Attributes | No. of Objects | No. of Classes |
|---|---|---|---|
| Breastcancer | 9 | 699 | 2 |
| BreastEW | 30 | 569 | 2 |
| Exactly | 13 | 1000 | 2 |
| HeartEW | 13 | 270 | 2 |
| Lymphography | 18 | 148 | 4 |
| PenglungEW | 325 | 73 | 7 |
| SonarEW | 60 | 208 | 2 |
| SpectEW | 22 | 267 | 2 |
| CongressEW | 16 | 435 | 2 |
| IonosphereEW | 34 | 351 | 2 |
| KrvskpEW | 36 | 3196 | 2 |
| WaveformEW | 40 | 5000 | 3 |
| WineEW | 13 | 178 | 3 |
| Zoo | 16 | 101 | 7 |

*4.2. Parameter Settings*

In this study, K-nearest neighbors (KNN) was utilized as the classification algorithm to evaluate the accuracy of the selected features. The datasets were divided into training and test sets using 10-fold cross-validation, and the classification error was calculated using K-nearest neighbors (KNN) with $k = 5$. This experiment was conducted on a WIN10 system with NVIDIA GTX 1660 graphics card, Inter Core i5-11400 processor, 2.6 GHz main frequency, and 16 GB RAM; the 2021a version of matlab was used. To ensure the fairness of the experiments, we used the same parameter settings and biomimetic environment as described in the original paper. The relevant parameters of the algorithm are set as shown in Table 2 below:

**Table 2.** The parameter settings of algorithms.

| Parameter | Value |
|---|---|
| Population size | 10 |
| The maximum number of iterations | 100 |
| $K$ parameters in KNN | 5 |
| $CR$ | 0.3 |
| $\beta$ of SBDA | $\frac{\pi}{3}$ |
| $a$ of BGWO | from 2 to 0 |
| $G_0$ of BGSA | 100 |
| $\alpha$ of BGSA | 20 |

*4.3. Comparison with LBDA Method*

In this section, we obtain results on 14 mainstream datasets using the proposed FS method. The obtained results are compared with LBDA in three aspects: classification accuracy, number of selected features and fitness value. Table 3 shows the comparison between BDA-DDO and LBDA in terms of classification accuracy, and it should be noted that LBDA1 indicates that the proposed algorithm does not apply an adaptive step-updating mechanism. Table 4 shows the comparison of the proposed algorithm with LBDA in selecting the number of features. Table 5 shows that BDA-DDO compares with LBDA in terms of fitness value.



**Table 3.** Average classification accuracy based on the proposed LBDA-DDO algorithm.

| DataSet | Metric | LBDA | LBDA1 | LBDA-DDO |
|---|---|---|---|---|
| Breastcancer | Avg | **0.999** | **0.999** | **0.999** |
| | StDev | 0.003 | 0.002 | 0.003 |
| BreastEW | Avg | 0.993 | **0.994** | **0.994** |
| | StDev | 0.004 | 0.004 | 0.004 |
| Exactly | Avg | 0.952 | 0.972 | **1** |
| | StDev | 0.053 | 0.033 | 0 |
| HeartEW | Avg | 0.903 | 0.921 | **0.935** |
| | StDev | 0.020 | 0.018 | 0.019 |
| Lymphography | Avg | 0.956 | **0.970** | 0.964 |
| | StDev | 0.019 | 0.012 | 0.014 |
| PenglungEW | Avg | **1** | **1** | **1** |
| | StDev | 0 | 0 | 0 |
| SonarEW | Avg | 0.954 | 0.960 | **0.964** |
| | StDev | 0.019 | 0.013 | 0.015 |
| SpectEW | Avg | 0.922 | **0.929** | **0.929** |
| | StDev | 0.014 | 0.014 | 0.017 |
| CongressEW | Avg | 0.998 | 0.999 | **1** |
| | StDev | 0.003 | 0.004 | 0 |
| IonosphereEW | Avg | 0.960 | 0.966 | **0.967** |
| | StDev | 0.008 | 0.011 | 0.011 |
| KrvskpEW | Avg | 0.970 | 0.968 | **0.985** |
| | StDev | 0.007 | 0.005 | 0.003 |
| WaveformEW | Avg | 0.831 | 0.829 | **0.845** |
| | StDev | 0.005 | 0.005 | 0.005 |
| WineEW | Avg | **1** | **1** | **1** |
| | StDev | 0 | 0 | 0 |
| Zoo | Avg | **1** | **1** | **1** |
| | StDev | 0 | 0 | 0 |

**Table 4.** Average selected features based on proposed LBDA-DDO algorithm.

| DataSet | Metric | LBDA | LBDA1 | LBDA-DDO |
|---|---|---|---|---|
| Breastcancer | Avg | 4.97 | 4.57 | **4.46** |
| | StDev | 1.30 | 1.22 | 1.20 |
| BreastEW | Avg | 14.17 | 13.93 | **12.57** |
| | StDev | 2.07 | 2.82 | 2.01 |
| Exactly | Avg | 7.30 | 7.17 | **6** |
| | StDev | 0.78 | 0.58 | 0 |
| HeartEW | Avg | 5.70 | 4.77 | **4.33** |
| | StDev | 1.24 | 1.54 | 1.42 |
| Lymphography | Avg | 7.73 | 8.13 | **6.77** |
| | StDev | 2.06 | 2.44 | 2.06 |
| PenglungEW | Avg | 126.13 | 106.13 | **105.30** |
| | StDev | 19.17 | 5.93 | 9.54 |
| SonarEW | Avg | 27.63 | 24.43 | **23.00** |
| | StDev | 3.66 | 6.00 | 3.85 |

**Table 4.** *Cont.*

| DataSet | Metric | LBDA | LBDA1 | LBDA-DDO |
|---------|--------|------|-------|----------|
| SpectEW | Avg | 9.83 | 9.50 | **8.07** |
| | StDev | 2.65 | 4.09 | 3.10 |
| CongressEW | Avg | 6.50 | 4.83 | **4.33** |
| | StDev | 2.19 | 1.80 | 1.81 |
| IonosphereEW | Avg | 15.33 | 12.80 | **11.17** |
| | StDev | 3.27 | 3.67 | 3.10 |
| KrvskpEW | Avg | 19.90 | 19.07 | **16.53** |
| | StDev | 2.75 | 3.08 | 2.30 |
| WaveformEW | Avg | 22.37 | 22.67 | **19.83** |
| | StDev | 3.11 | 3.38 | 2.55 |
| WineEW | Avg | 4.43 | 3.70 | **2.67** |
| | StDev | 0.88 | 0.82 | 0.65 |
| Zoo | Avg | 4.90 | 4.40 | **3.93** |
| | StDev | 0.70 | 0.66 | 0.72 |

**Table 5.** Average fitness based on the proposed LBDA-DDO algorithm.

| DataSet | Metric | LBDA | LBDA1 | LBDA-DDO |
|---------|--------|------|-------|----------|
| Breastcancer | Avg | 0.007 | **0.006** | **0.006** |
| | StDev | 0.003 | 0.002 | 0.002 |
| BreastEW | Avg | 0.011 | 0.010 | **0.009** |
| | StDev | 0.004 | 0.004 | 0.004 |
| Exactly | Avg | 0.052 | 0.032 | **0.005** |
| | StDev | 0.053 | 0.058 | 0 |
| HeartEW | Avg | 0.101 | 0.081 | **0.066** |
| | StDev | 0.020 | 0.017 | 0.019 |
| Lymphography | Avg | 0.047 | **0.034** | 0.039 |
| | StDev | 0.019 | 0.011 | 0.014 |
| PenglungEW | Avg | 0.004 | **0.003** | **0.003** |
| | StDev | 0.001 | 0 | 0 |
| SonarEW | Avg | 0.050 | 0.044 | **0.039** |
| | StDev | 0.018 | 0.012 | 0.014 |
| SpectEW | Avg | 0.082 | 0.075 | **0.073** |
| | StDev | 0.013 | 0.013 | 0.017 |
| CongressEW | Avg | 0.006 | 0.004 | **0.002** |
| | StDev | 0.004 | 0.003 | 0.001 |
| IonosphereEW | Avg | 0.044 | 0.038 | **0.036** |
| | StDev | 0.008 | 0.010 | 0.011 |
| KrvskpEW | Avg | 0.036 | 0.037 | **0.019** |
| | StDev | 0.007 | 0.005 | 0.003 |
| WaveformEW | Avg | 0.173 | 0.175 | **0.157** |
| | StDev | 0.005 | 0.005 | 0.005 |
| WineEW | Avg | 0.003 | 0.003 | **0.002** |
| | StDev | 0.001 | 0.001 | 0 |
| Zoo | Avg | 0.003 | 0.003 | **0.002** |
| | StDev | 0 | 0 | 0 |

Table 3 shows the classification accuracy obtained by LBDA-DDO, LBDA1, and LBDA on the 14 datasets. The results indicate that LBDA1 achieves higher classification accuracy on most datasets compared to LBDA, while LBDA-DDO consistently outperforms LBDA in terms of classification accuracy on all 14 datasets. Furthermore, considering the standard deviations, LBDA-DDO demonstrates better robustness than LBDA, indicating its ability to handle variations in the datasets more effectively. Therefore, LBDA-DDO can achieve better results by utilizing an adaptive step size strategy, which enhances the algorithm's search capability in the later stages and balances the trade-off between exploration and exploitation.

Table 4 illustrates the average number of selected features by the proposed LBDA-DDO algorithm on the datasets, with the best results highlighted in bold. The findings in Table 4 indicate that LBDA-DDO consistently selects fewer features than LBDA and LBDA1 across all 14 datasets. Furthermore, LBDA1 demonstrates a lower feature count compared to LBDA. These results demonstrate the ability of the proposed algorithm to effectively reduce the number of selected features in comparison to LBDA, thereby eliminating noisy or irrelevant features that may have been chosen by the LBDA method. This outcome aligns perfectly with our main objective.

Table 5 illustrates the average fitness results obtained by the proposed LBDA-DDO algorithm. As before, the best results are shown in bold. The results show that the fitness values obtained by LBDA-DDO on the nine datasets are lower than those of LBDA and LBDA1. On the two datasets Breastcancer and penglungEW, LBDA-DDO and LBDA1 exhibit the same fitness values. On Lymphography, LBDA1 achieves the best results. Overall, LBDA-DDO performs better than LBDA.

Figure 1 presents the convergence behavior of the proposed LBDA-DDO algorithm on 14 mainstream datasets, comparing it with LBDA and LBDA1. LBDA refers to the binary dragonfly algorithm with linear model updating, while LBDA1 represents the proposed algorithm(LBDA-DDO) without the adaptive step-updating mechanism. The *X*-axis shows the number of iterations, and the *Y*-axis displays the fitness value. The results show that the proposed algorithm achieves better convergence on most datasets. This indicates an enhanced search capability, especially with the inclusion of the adaptive step-updating mechanism. The proposed algorithm effectively avoids getting stuck in local optima.

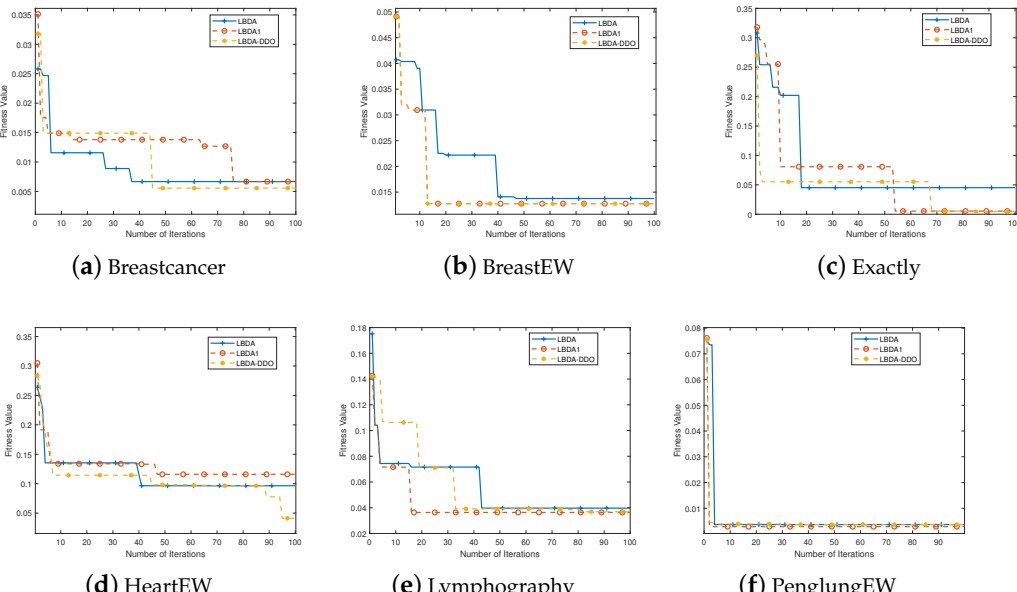

**(a)** Breastcancer  **(b)** BreastEW  **(c)** Exactly

**(d)** HeartEW  **(e)** Lymphography  **(f)** PenglungEW

**Figure 1.** *Cont.*

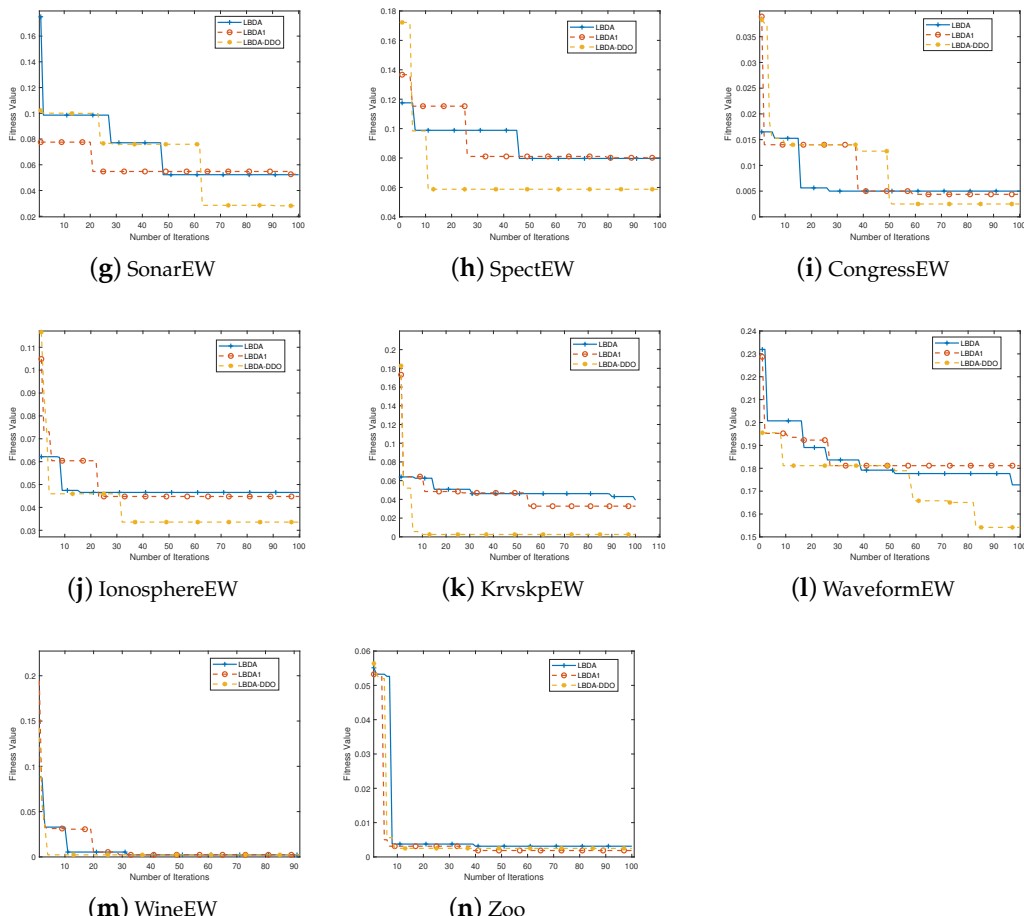

**Figure 1.** Convergence behavior of the proposed algorithm based on LBDA on 14 mainstream datasets.

Additionally, it can be observed that LBDA exhibits slower convergence on certain datasets, necessitating more iterations to attain satisfactory fitness values. Consequently, the proposed algorithm holds an advantage in terms of convergence behavior, enabling it to swiftly discover high-quality solutions. This highlights the efficacy of the introduced directional difference operator strategy in accelerating convergence speed, while the adaptive step-updating mechanism enhances search capabilities. In summary, the proposed algorithm outperforms LBDA by achieving faster convergence and yielding solutions with higher fitness values.

### 4.4. Comparison with QBDA Method

In this section, the performance of the proposed QBDA-DDO algorithm is examined, which combines the QBDA method with DE. QBDA1 is also included as an algorithm without the adaptive step size mechanism. For the performance of the algorithm, a comparison is conducted with QBDA in three aspects: classification accuracy, selected features, and fitness values. Table 6 presents the comparison of the classification accuracy between the proposed algorithm and QBDA. Table 7 shows the comparison of selected features between QBDA-DDO and QBDA. Additionally, Table 8 provides a comparison of fitness values between the QBDA-DDO algorithm and QBDA. Furthermore, Figure 2 shows the convergence speed of the three algorithms on the 14 mainstream datasets for visual analysis.

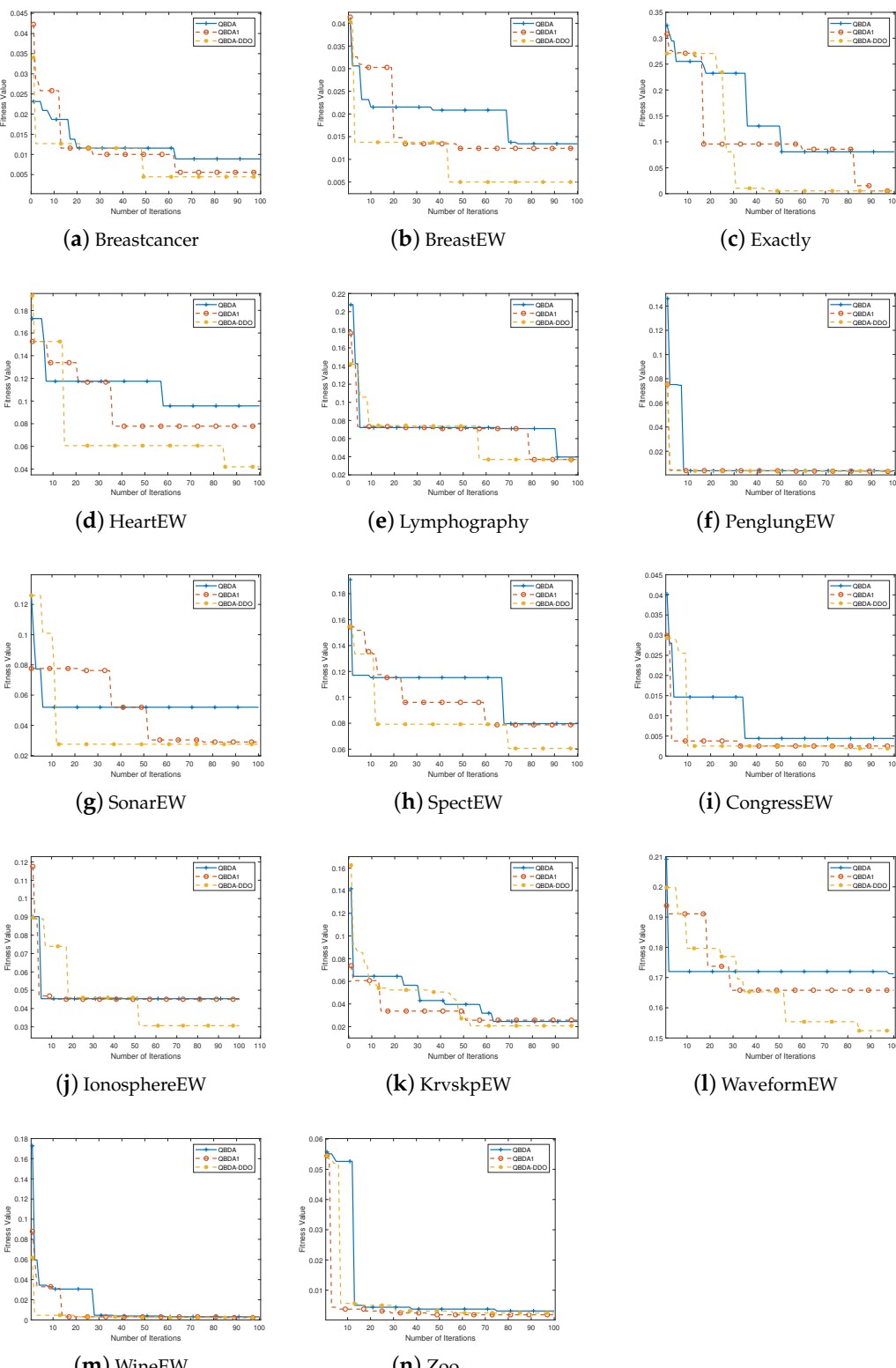

**Figure 2.** Convergence behavior of the proposed algorithm based on QBDA on 14 mainstream datasets.

**Table 6.** The average classification accuracy of the proposed algorithm QBDA-DDO.

| DataSet | Metric | QBDA | QBDA1 | QBDA-DDO |
|---|---|---|---|---|
| Breastcancer | Avg | 0.998 | **0.999** | **0.999** |
| | StDev | 0.002 | 0.002 | 0.002 |
| BreastEW | Avg | 0.994 | **0.995** | 0.994 |
| | StDev | 0.004 | 0.004 | 0.004 |
| Exactly | Avg | 0.974 | 0.990 | **1** |
| | StDev | 0.044 | 0.023 | 0 |
| HeartEW | Avg | 0.915 | 0.923 | **0.945** |
| | StDev | 0.017 | 0.018 | 0.019 |
| Lymphography | Avg | 0.963 | 0.969 | **0.971** |
| | StDev | 0.023 | 0.018 | 0.020 |
| PenglungEW | Avg | **1** | **1** | **1** |
| | StDev | 0 | 0 | 0 |
| SonarEW | Avg | 0.958 | **0.968** | 0.965 |
| | StDev | 0.015 | 0.017 | 0.016 |
| SpectEW | Avg | 0.920 | 0.930 | **0.940** |
| | StDev | 0.014 | 0.013 | 0.013 |
| CongressEW | Avg | 0.999 | **1** | **1** |
| | StDev | 0.002 | 0.002 | 0 |
| IonosphereEW | Avg | 0.959 | 0.965 | **0.970** |
| | StDev | 0.009 | 0.011 | 0.011 |
| KrvskpEW | Avg | 0.972 | 0.975 | **0.984** |
| | StDev | 0.004 | 0.005 | 0.005 |
| WaveformEW | Avg | 0.834 | 0.835 | **0.849** |
| | StDev | 0.006 | 0.004 | 0.010 |
| WineEW | Avg | **1** | **1** | **1** |
| | StDev | 0 | 0 | 0 |
| Zoo | Avg | **1** | **1** | **1** |
| | StDev | 0 | 0 | 0 |

**Table 7.** The average number of the selected features of the proposed algorithm QBDA-DDO.

| DataSet | Metric | QBDA | QBDA1 | QBDA-DDO |
|---|---|---|---|---|
| Breastcancer | Avg | 4.80 | 4.57 | **4.13** |
| | StDev | 1.22 | 1.17 | 0.76 |
| BreastEW | Avg | 14.73 | 13.73 | **11.77** |
| | StDev | 2.43 | 2.95 | 1.83 |
| Exactly | Avg | 7.00 | 6.63 | **6.03** |
| | StDev | 0.89 | 0.60 | 0.17 |
| HeartEW | Avg | 5.9 | **4.53** | **4.53** |
| | StDev | 1.85 | 1.82 | 1.17 |
| Lymphography | Avg | 8.37 | 7.30 | **7.03** |
| | StDev | 2.23 | 2.18 | 1.74 |
| PenglungEW | Avg | 128.00 | 112.17 | **108.90** |
| | StDev | 6.67 | 5.01 | 5.73 |
| SonarEW | Avg | 27.13 | 23.00 | **22.60** |
| | StDev | 3.30 | 4.60 | 3.92 |

**Table 7.** *Cont.*

| DataSet | Metric | QBDA | QBDA1 | QBDA-DDO |
|---|---|---|---|---|
| SpectEW | Avg | 9.00 | 10.67 | **7.97** |
| | StDev | 1.71 | 2.87 | 2.96 |
| CongressEW | Avg | 6.17 | 5.07 | **4.10** |
| | StDev | 1.97 | 1.36 | 1.12 |
| IonosphereEW | Avg | 14.00 | 12.40 | **10.37** |
| | StDev | 3.29 | 4.02 | 3.02 |
| KrvskpEW | Avg | 19.67 | 19.33 | **17.90** |
| | StDev | 2.38 | 3.05 | 2.00 |
| WaveformEW | Avg | 22.87 | 21.83 | **21.20** |
| | StDev | 2.28 | 2.50 | 3.15 |
| WineEW | Avg | 4.53 | 3.63 | **3.06** |
| | StDev | 1.15 | 0.66 | 0.51 |
| Zoo | Avg | 5.03 | 4.23 | **3.87** |
| | StDev | 0.91 | 0.67 | 0.42 |

**Table 8.** The average fitness value of the proposed algorithm QBDA-DDO.

| DataSet | Metric | QBDA | QBDA1 | QBDA-DDO |
|---|---|---|---|---|
| Breastcancer | Avg | 0.006 | 0.006 | **0.005** |
| | StDev | 0.002 | 0.002 | 0.002 |
| BreastEW | Avg | 0.010 | **0.009** | **0.009** |
| | StDev | 0.004 | 0.004 | 0.004 |
| Exactly | Avg | 0.031 | 0.015 | **0.005** |
| | StDev | 0.044 | 0.024 | 0 |
| HeartEW | Avg | 0.088 | 0.079 | **0.058** |
| | StDev | 0.016 | 0.017 | 0.019 |
| Lymphography | Avg | 0.041 | 0.035 | **0.032** |
| | StDev | 0.022 | 0.017 | 0.020 |
| PenglungEW | Avg | 0.004 | **0.003** | **0.003** |
| | StDev | 0.002 | 0 | 0 |
| SonarEW | Avg | 0.045 | **0.035** | 0.038 |
| | StDev | 0.015 | 0.017 | 0.016 |
| SpectEW | Avg | 0.082 | 0.073 | **0.063** |
| | StDev | 0.014 | 0.013 | 0.013 |
| CongressEW | Avg | 0.004 | 0.003 | **0.002** |
| | StDev | 0.002 | 0.002 | 0 |
| IonosphereEW | Avg | 0.044 | 0.038 | **0.032** |
| | StDev | 0.009 | 0.010 | 0.012 |
| KrvskpEW | Avg | 0.033 | 0.030 | **0.020** |
| | StDev | 0.004 | 0.005 | 0.005 |
| WaveformEW | Avg | 0.170 | 0.169 | **0.154** |
| | StDev | 0.006 | 0.004 | 0.010 |
| WineEW | Avg | 0.003 | 0.003 | **0.002** |
| | StDev | 0 | 0 | 0 |
| Zoo | Avg | 0.003 | 0.003 | **0.002** |
| | StDev | 0 | 0 | 0 |

Table 6 displays the average classification accuracy achieved by the proposed QBDA-DDO algorithm. QBDA-DDO achieves the best results on seven datasets, while the results of the three algorithms are equal on three datasets. QBDA1 and BDA-DDO obtain the same results on Breastcancer and CongressEW. For BreastEW and SonarEW, QBDA1 performs the best. Therefore, the directional differential operator and adaptive step size, when applied to QBDA, can effectively improve the classification accuracy of feature selection, highlighting the effectiveness of the algorithm innovation.

Table 7 shows the average number of selected features obtained by the proposed BDA-DDO algorithm, with the best results highlighted in bold. The results demonstrate that the BDA-DDO algorithm achieves the best results on all 13 datasets, while QBDA1 outperforms QBDA on most datasets. This indicates that the directional differential operator mechanism assists QBDA in converging to favorable solutions, while the adaptive step size enhances the algorithm's search capability in the later stages. In summary, these improvements facilitate the removal of redundant and noisy features, resulting in enhancing algorithm performance.

Table 8 presents the average fitness values obtained by the QBDA-DDO algorithm, with the best results highlighted in bold. It is evident from the table that QBDA-DDO consistently outperforms QBDA in terms of average fitness values on the majority of datasets. The fitness value serves as an indicator of the combined performance in terms of classification accuracy and the number of selected features. Thus, the proposed algorithm demonstrates superior performance compared to QBDA, showcasing its effectiveness in better overall results.

Figure 2 depicts the convergence behavior of the proposed QBDA-DDO algorithm compared to QBDA and QBDA1 on 14 mainstream datasets. QBDA represents the binary dragonfly algorithm with quadratic model-based parameter updates. The *X*-axis represents the number of iterations, and the *Y*-axis represents the fitness values. The results demonstrate that the proposed QBDA-DDO algorithm exhibits superior search capability in the later stages compared to QBDA for most datasets. The proposed algorithm effectively avoids getting trapped in local optima.

*4.5. Comparison with SBDA Method*

In this section, we carried out tests on the proposed method from three perspectives: classification accuracy, number of selected features, and fitness value. We also compared it with the SBDA algorithm, where SBDA1 represents the algorithm without the application of an adaptive step size mechanism. Table 9 presents the classification accuracy achieved by the proposed method on the 14 mainstream datasets. Table 10 displays the number of selected features obtained by SBDA-DDO. Table 11 shows the fitness values achieved by SBDA-DDO on the 14 datasets. Additionally, Figure 3 is used to compare the convergence speed of SBDA-DDO and SBDA on the 14 mainstream datasets.

In terms of classification accuracy, according to the results in Table 9, the SBDA-DDO algorithm achieves the best results on seven datasets, which are highlighted in bold. It is noteworthy that SBDA1 achieves the same classification accuracy as QBDA-DDO on five datasets, and all three algorithms show the same accuracy on three datasets. This indicates that the integration of the direction differential operator and the adaptive step size mechanism further improves the performance of the SBDA algorithm. Therefore, SBDA-DDO demonstrates superior ability to classify the datasets.

In terms of the number of selected features, it can be seen from Table 10 that BDA-DDO can achieve better performance, BDA-DDO achieves the best results on eight datasets, and SBDA1 obtained the best results on six datasets. Therefore, BDA-DDO is able to select fewer features, proving that the adaptive step can improve the performance of the algorithm, while the results obtained by SBDA1 are better than the original SBDA algorithm.

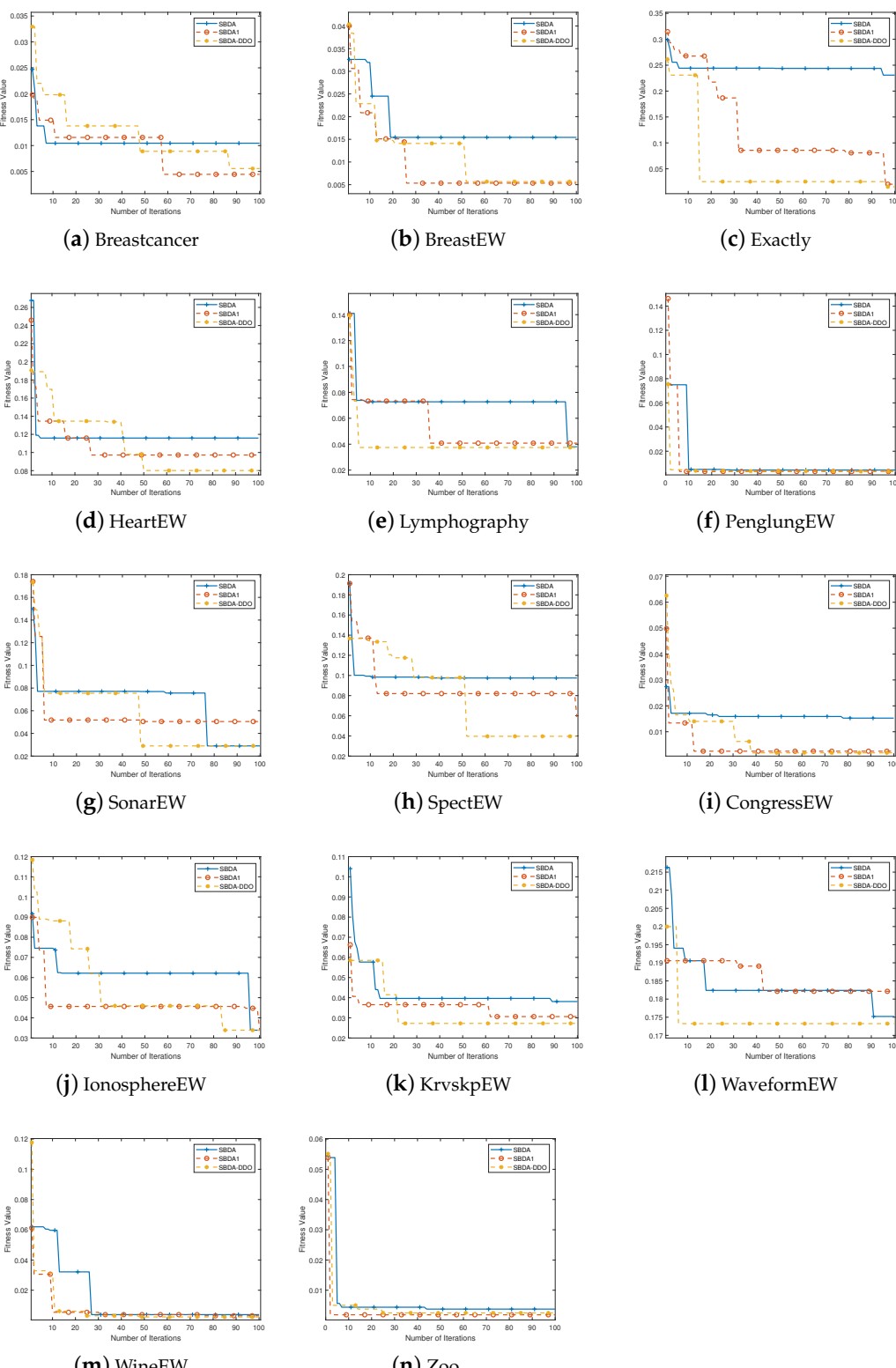

**Figure 3.** Convergence behavior of the proposed algorithm based on SBDA on 14 mainstream datasets.

**Table 9.** The average classification accuracy of the proposed algorithm SBDA-DDO.

| DataSet | Metric | SBDA | SBDA1 | SBDA-DDO |
|---------|--------|------|-------|----------|
| Breastcancer | Avg | 0.998 | **0.999** | **0.999** |
| | StDev | 0.003 | 0.003 | 0.002 |
| BreastEW | Avg | 0.993 | **0.994** | 0.993 |
| | StDev | 0.003 | 0.004 | 0.004 |
| Exactly | Avg | 0.946 | 0.960 | **0.987** |
| | StDev | 0.060 | 0.052 | 0.025 |
| HeartEW | Avg | 0.897 | 0.909 | **0.919** |
| | StDev | 0.018 | 0.020 | 0.019 |
| Lymphography | Avg | **1** | **1** | **1** |
| | StDev | 0 | 0 | 0 |
| PenglungEW | Avg | **1** | **1** | **1** |
| | StDev | 0 | 0 | 0 |
| SonarEW | Avg | 0.960 | 0.963 | **0.964** |
| | StDev | 0.013 | 0.020 | 0.015 |
| SpectEW | Avg | 0.924 | **0.929** | **0.929** |
| | StDev | 0.018 | 0.014 | 0.018 |
| CongressEW | Avg | 0.997 | 0.999 | **1** |
| | StDev | 0.005 | 0.003 | 0 |
| IonosphereEW | Avg | 0.959 | 0.961 | **0.968** |
| | StDev | 0.010 | 0.011 | 0.011 |
| KrvskpEW | Avg | 0.968 | 0.965 | **0.974** |
| | StDev | 0.005 | 0.005 | 0.004 |
| WaveformEW | Avg | 0.832 | 0.828 | **0.834** |
| | StDev | 0.006 | 0.007 | 0.006 |
| WineEW | Avg | **1** | **1** | **1** |
| | StDev | 0 | 0 | 0 |
| Zoo | Avg | **1** | **1** | **1** |
| | StDev | 0 | 0 | 0 |

**Table 10.** The average number of the selected features of the proposed algorithm SBDA-DDO.

| DataSet | Metric | SBDA | SBDA1 | SBDA-DDO |
|---------|--------|------|-------|----------|
| Breastcancer | Avg | 4.90 | **4.03** | 5.07 |
| | StDev | 1.32 | 1.05 | 1.34 |
| BreastEW | Avg | 14.40 | 13.63 | **13.17** |
| | StDev | 2.39 | 3.73 | 2.74 |
| Exactly | Avg | 7.47 | 7.13 | **6.77** |
| | StDev | 0.62 | 0.80 | 0.61 |
| HeartEW | Avg | 4.87 | **4.13** | 4.80 |
| | StDev | 1.23 | 1.17 | 1.25 |
| Lymphography | Avg | 8.77 | 7.97 | **7.73** |
| | StDev | 2.17 | 2.64 | 2.00 |
| PenglungEW | Avg | 136.50 | **103.03** | 108.47 |
| | StDev | 15.32 | 4.98 | 4.72 |
| SonarEW | Avg | 28.93 | 24.30 | **23.73** |
| | StDev | 3.95 | 6.46 | 5.11 |

**Table 10.** *Cont.*

| DataSet | Metric | SBDA | SBDA1 | SBDA-DDO |
|---------|--------|------|-------|----------|
| SpectEW | Avg | 10.50 | **8.57** | 9.00 |
| | StDev | 2.56 | 3.80 | 3.63 |
| CongressEW | Avg | 6.63 | **4.43** | 5.10 |
| | StDev | 2.55 | 1.41 | 1.87 |
| IonosphereEW | Avg | 13.80 | **11.17** | 12.37 |
| | StDev | 2.26 | 2.58 | 3.65 |
| KrvskpEW | Avg | 20.07 | 19.47 | **18.63** |
| | StDev | 2.05 | 3.87 | 2.73 |
| WaveformEW | Avg | 22.97 | 21.43 | **20.50** |
| | StDev | 3.17 | 3.57 | 2.80 |
| WineEW | Avg | 4.70 | 3.60 | **3.23** |
| | StDev | 1.29 | 0.88 | 0.67 |
| Zoo | Avg | 5.13 | 4.50 | **4.13** |
| | StDev | 0.76 | 0.88 | 0.88 |

**Table 11.** The average fitness value of the proposed algorithm SBDA-DDO.

| DataSet | Metric | SBDA | SBDA1 | SBDA-DDO |
|---------|--------|------|-------|----------|
| Breastcancer | Avg | 0.007 | **0.006** | **0.006** |
| | StDev | 0.002 | 0.002 | 0.002 |
| BreastEW | Avg | 0.012 | **0.011** | **0.011** |
| | StDev | 0.003 | 0.004 | 0.004 |
| Exactly | Avg | 0.059 | 0.045 | **0.018** |
| | StDev | 0.060 | 0.053 | 0.025 |
| HeartEW | Avg | 0.105 | 0.093 | **0.083** |
| | StDev | 0.018 | 0.020 | 0.018 |
| Lymphography | Avg | 0.046 | **0.038** | 0.042 |
| | StDev | 0.016 | 0.021 | 0.018 |
| PenglungEW | Avg | 0.004 | **0.003** | **0.003** |
| | StDev | 0 | 0 | 0 |
| SonarEW | Avg | 0.047 | 0.040 | **0.039** |
| | StDev | 0.013 | 0.020 | 0.014 |
| SpectEW | Avg | 0.079 | **0.074** | **0.074** |
| | StDev | 0.018 | 0.014 | 0.017 |
| CongressEW | Avg | 0.006 | **0.003** | **0.003** |
| | StDev | 0.004 | 0.002 | 0.001 |
| IonosphereEW | Avg | 0.045 | 0.041 | **0.036** |
| | StDev | 0.010 | 0.011 | 0.010 |
| KrvskpEW | Avg | 0.037 | 0.040 | **0.031** |
| | StDev | 0.005 | 0.005 | 0.004 |
| WaveformEW | Avg | 0.172 | 0.176 | **0.169** |
| | StDev | 0.006 | 0.007 | 0.006 |
| WineEW | Avg | 0.004 | 0.003 | **0.002** |
| | StDev | 0 | 0 | 0 |
| Zoo | Avg | 0.003 | 0.003 | **0.002** |
| | StDev | 0 | 0 | 0 |

Table 11 illustrates the results obtained by the proposed SBDA-DDO algorithm in terms of fitness values, with the best results highlighted in bold. It can be observed that

SBDA-DDO outperforms other algorithms on 13 datasets, while SBDA1 achieves the best results on 6 datasets. Remarkably, SBDA-DDO and SBDA1 yield the same results on five datasets. These findings confirm the effectiveness of our innovative approach, which incorporates direction differential operators and adaptive step size mechanisms into three different methods. The combination of these innovations with each algorithm demonstrates superior performance compared to the original algorithms. Consequently, our proposed approach enhances the algorithm's performance by achieving improved classification accuracy while selecting fewer features.

Figure 3 depicts the convergence behavior of the proposed SBDA-DDO algorithm on 14 datasets, and the comparison with SBDA and SBDA1. SBDA represents the binary dragonfly algorithm with parameter updates using the cosine model. The *X*-axis represents the number of iterations, and the *Y*-axis displays the corresponding fitness values. Results show that SBDA-DDO exhibits faster convergence than SBDA for most datasets, thanks to the direction differential operator guiding individuals towards more promising directions. In addition, the later search ability of this algorithm is obviously stronger than that of SBDA, especially after using the adaptive step size mechanism.

The result of Figures 1–3 confirms the significant impact of the proposed innovative mechanisms on enhancing the performance of BDA with various update strategies (linear, quadratic, and cosine). The results indicate that the introduced improvements effectively boost the performance of the three BDA variants, particularly improving their search capabilities in later stages and facilitating escape from local optima. This highlights the versatility and effectiveness of the proposed mechanisms in various scenarios, affirming the algorithm's robustness and adaptability.

### 4.6. Comparison with Other Binary Optimization Algorithms

In the previous section, we compared the proposed BDA-DDO algorithm and three enhanced versions of BDA (LBDA, QBDA, SBDA) [40] in terms of classification accuracy, selected feature count, and fitness values. Now, we extend the comparison to four other typical metaheuristic feature selection algorithms: binary grey wolf optimization (BGWO) [48], the binary bat algorithm (BBA) [49], binary atomic search optimization (BASO) [50], and the binary gravitational search algorithm (BGSA) [49]. The results are presented in Tables 12–14 for classification accuracy, selected feature count, and fitness values, respectively. By considering these additional algorithms, we aim to provide a comprehensive evaluation of the performance of the proposed BDA-DDO algorithm.

**Table 12.** Comparison with other binary algorithms on average classification accuracy.

| DataSet | Metric | LBDA-DDO | QBDA-DDO | SBDA-DDO | BGWO | BBA | BASO | BGSA |
|---------|--------|----------|----------|----------|------|-----|------|------|
| Breastcancer | Avg | **0.999** | **0.999** | **0.999** | 0.978 | 0.980 | 0.971 | 0.948 |
| | StDev | 0.003 | 0.002 | 0.002 | 0.011 | 0.009 | 0.013 | 0.051 |
| BreastEW | Avg | **0.994** | **0.994** | 0.993 | 0.978 | 0.983 | 0.960 | 0.928 |
| | StDev | 0.004 | 0.004 | 0.004 | 0.013 | 0.009 | 0.018 | 0.014 |
| Exactly | Avg | **1** | **1** | 0.987 | 0.805 | 0.982 | 0.727 | 0.732 |
| | StDev | 0 | 0 | 0.025 | 0.065 | 0.069 | 0.070 | 0.124 |
| HeartEW | Avg | 0.935 | **0.945** | 0.919 | 0.836 | 0.887 | 0.825 | 0.770 |
| | StDev | 0.019 | 0.019 | 0.019 | 0.057 | 0.035 | 0.042 | 0.066 |
| Lymphography | Avg | 0.964 | 0.971 | **1** | 0.872 | 0.911 | 0.859 | 0.864 |
| | StDev | 0.014 | 0.020 | 0 | 0.053 | 0.049 | 0.049 | 0.081 |
| PenglungEW | Avg | **1** | **1** | **1** | 0.883 | 0.924 | 0.924 | 0.949 |
| | StDev | 0 | 0 | 0 | 0.072 | 0.069 | 0.074 | 0.054 |
| SonarEW | Avg | 0.964 | **0.965** | 0.964 | 0.902 | 0.939 | 0.881 | 0.865 |
| | StDev | 0.015 | 0.016 | 0.015 | 0.044 | 0.039 | 0.049 | 0.047 |
| SpectEW | Avg | 0.929 | **0.940** | 0.929 | 0.863 | 0.894 | 0.831 | 0.785 |
| | StDev | 0.017 | 0.013 | 0.018 | 0.044 | 0.036 | 0.031 | 0.034 |

**Table 12.** *Cont.*

| DataSet | Metric | LBDA-DDO | QBDA-DDO | SBDA-DDO | BGWO | BBA | BASO | BGSA |
|---------|--------|----------|----------|----------|------|-----|------|------|
| CongressEW | Avg | **1** | **1** | **1** | 0.977 | 0.983 | 0.963 | 0.943 |
| | StDev | 0 | 0 | 0 | 0.018 | 0.012 | 0.023 | 0.026 |
| IonosphereEW | Avg | 0.967 | **0.970** | 0.968 | 0.908 | 0.932 | 0.909 | 0.869 |
| | StDev | 0.011 | 0.011 | 0.011 | 0.037 | 0.025 | 0.025 | 0.026 |
| KrvskpEW | Avg | **0.985** | 0.984 | 0.974 | 0.972 | 0.983 | 0.904 | 0.898 |
| | StDev | 0.003 | 0.005 | 0.004 | 0.008 | 0.005 | 0.041 | 0.051 |
| WaveformEW | Avg | 0.845 | **0.849** | 0.834 | 0.843 | 0.845 | 0.799 | 0.697 |
| | StDev | 0.005 | 0.010 | 0.006 | 0.011 | 0.010 | 0.015 | 0.021 |
| WineEW | Avg | **1** | **1** | **1** | 0.955 | 0.986 | 0.946 | 0.976 |
| | StDev | 0 | 0 | 0 | 0.045 | 0.016 | 0.049 | 0.035 |
| Zoo | Avg | **1** | **1** | **1** | 0.943 | 0.985 | 0.933 | 0.995 |
| | StDev | 0 | 0 | 0 | 0.059 | 0.026 | 0.045 | 0.015 |

Based on the results in Table 12, we compared our proposed method with BGWO, BBA, BASO, and BGSA in terms of classification accuracy. The best results are highlighted in bold. QBDA-DDO achieved the highest classification accuracy on 12 datasets, while LBDA-DDO and SBDA-DDO obtained the best results on 8 and 6 datasets, respectively. Therefore, QBDA outperformed other algorithms in improving classification accuracy.

**Table 13.** Comparison with other binary algorithms on average number of selected features.

| DataSet | Metric | LBDA-DDO | QBDA-DDO | SBDA-DDO | BGWO | BBA | BASO | BGSA |
|---------|--------|----------|----------|----------|------|-----|------|------|
| Breastcancer | Avg | 4.46 | **4.13** | 5.07 | 5.23 | 4.17 | 4.23 | 4.47 |
| | StDev | 1.20 | 0.76 | 1.34 | 1.45 | 1.21 | 1.23 | 1.01 |
| BreastEW | Avg | 12.57 | 11.77 | 13.17 | 17.93 | **11.33** | 11.90 | 14.93 |
| | StDev | 2.01 | 1.83 | 2.74 | 2.68 | 2.28 | 3.40 | 2 |
| Exactly | Avg | 6 | 6.03 | 6.77 | 9.97 | 6 | **5.57** | 7.67 |
| | StDev | 0 | 0.17 | 0.61 | 1.54 | 0.45 | 3.29 | 1.49 |
| HeartEW | Avg | **4.33** | 4.53 | 4.80 | 6.97 | 4.50 | 4.40 | 6.63 |
| | StDev | 1.42 | 1.117 | 1.25 | 1.54 | 1.06 | 1.94 | 1.94 |
| Lymphography | Avg | 6.77 | 7.03 | 7.73 | 10.23 | **5.47** | 6.00 | 9 |
| | StDev | 2.06 | 1.74 | 2.00 | 1.71 | 1.45 | 2.21 | 2.18 |
| PenglungEW | Avg | 105.30 | 108.90 | 108.47 | 168.40 | 127.90 | **62.37** | 145.1 |
| | StDev | 9.54 | 5.73 | 4.72 | 21.79 | 8.33 | 32.81 | 4.88 |
| SonarEW | Avg | 23.00 | 22.60 | 23.73 | 33.93 | 23.27 | **18.27** | 27.07 |
| | StDev | 3.85 | 3.92 | 5.11 | 3.26 | 3.89 | 7.97 | 3.64 |
| SpectEW | Avg | 8.07 | **7.97** | 9.00 | 12.30 | 8.27 | 8.23 | 9.77 |
| | StDev | 3.10 | 2.96 | 3.63 | 3.00 | 1.86 | 4.07 | 2.3 |
| CongressEW | Avg | 4.33 | **4.10** | 5.10 | 8.03 | 5.33 | 4.17 | 7 |
| | StDev | 1.81 | 1.12 | 1.87 | 1.94 | 1.72 | 2.03 | 1.91 |
| IonosphereEW | Avg | 11.17 | 10.37 | 12.37 | 18.43 | 10.23 | **6.93** | 14.9 |
| | StDev | 3.10 | 3.02 | 3.65 | 2.55 | 2.58 | 1.93 | 2.89 |
| KrvskpEW | Avg | **16.53** | 17.90 | 18.63 | 28.40 | 19.80 | 18.27 | 19.73 |
| | StDev | 2.30 | 2.00 | 2.73 | 2.58 | 2.45 | 4.40 | 2.36 |
| WaveformEW | Avg | **19.83** | 21.20 | 20.50 | 31.27 | 22.93 | 21.27 | 21.6 |
| | StDev | 2.55 | 3.15 | 2.80 | 1.86 | 2.53 | 4.16 | 3.69 |
| WineEW | Avg | **2.67** | 3.06 | 3.23 | 6.53 | 3.53 | 3.73 | 6.57 |
| | StDev | 0.65 | 0.51 | 0.67 | 1.76 | 1.15 | 1.21 | 1.36 |
| Zoo | Avg | 3.93 | **3.87** | 4.13 | 8 | 5.37 | 5.53 | 6.97 |
| | StDev | 0.72 | 0.42 | 0.88 | 1.73 | 1.05 | 1.80 | 1.25 |

**Table 14.** Comparison with other binary-based algorithms in terms of average fitness value.

| DataSet | Metric | LBDA-DDO | QBDA-DDO | SBDA-DDO | BGWO | BBA | BASO | BGSA |
|---|---|---|---|---|---|---|---|---|
| Breastcancer | Avg | 0.006 | **0.005** | 0.006 | 0.027 | 0.024 | 0.033 | 0.027 |
| | StDev | 0.002 | 0.002 | 0.002 | 0.011 | 0.009 | 0.013 | 0.007 |
| BreastEW | Avg | **0.009** | **0.009** | 0.011 | 0.027 | 0.020 | 0.044 | 0.039 |
| | StDev | 0.004 | 0.004 | 0.004 | 0.012 | 0.009 | 0.018 | 0.01 |
| Exactly | Avg | **0.005** | **0.005** | 0.018 | 0.200 | 0.023 | 0.275 | 0.253 |
| | StDev | 0 | 0 | 0.025 | 0.065 | 0.068 | 0.069 | 0.094 |
| HeartEW | Avg | 0.066 | **0.058** | 0.083 | 0.167 | 0.115 | 0.176 | 0.137 |
| | StDev | 0.019 | 0.019 | 0.018 | 0.056 | 0.034 | 0.042 | 0.03 |
| Lymphography | Avg | 0.039 | **0.032** | 0.042 | 0.132 | 0.090 | 0.143 | 0.081 |
| | StDev | 0.014 | 0.020 | 0.018 | 0.052 | 0.049 | 0.048 | 0.033 |
| PenglungEW | Avg | **0.003** | **0.003** | **0.003** | 0.121 | 0.079 | 0.077 | 0.004 |
| | StDev | 0 | 0 | 0 | 0.072 | 0.068 | 0.073 | 0 |
| SonarEW | Avg | 0.039 | **0.038** | 0.039 | 0.102 | 0.064 | 0.120 | 0.082 |
| | StDev | 0.014 | 0.016 | 0.014 | 0.044 | 0.039 | 0.048 | 0.023 |
| SpectEW | Avg | 0.073 | **0.063** | 0.074 | 0.141 | 0.108 | 0.171 | 0.153 |
| | StDev | 0.017 | 0.013 | 0.017 | 0.043 | 0.035 | 0.030 | 0.018 |
| CongressEW | Avg | **0.002** | **0.002** | 0.003 | 0.027 | 0.020 | 0.039 | 0.032 |
| | StDev | 0.001 | 0 | 0.001 | 0.018 | 0.012 | 0.023 | 0.013 |
| IonosphereEW | Avg | 0.036 | **0.032** | 0.036 | 0.097 | 0.070 | 0.092 | 0.127 |
| | StDev | 0.011 | 0.012 | 0.010 | 0.037 | 0.025 | 0.024 | 0.011 |
| KrvskpEW | Avg | **0.019** | 0.020 | 0.031 | 0.035 | 0.022 | 0.100 | 0.099 |
| | StDev | 0.003 | 0.005 | 0.004 | 0.008 | 0.005 | 0.040 | 0.049 |
| WaveformEW | Avg | 0.157 | **0.154** | 0.169 | 0.163 | 0.159 | 0.204 | 0.251 |
| | StDev | 0.005 | 0.010 | 0.006 | 0.011 | 0.010 | 0.015 | 0.013 |
| WineEW | Avg | **0.002** | **0.002** | **0.002** | 0.049 | 0.017 | 0.057 | 0.009 |
| | StDev | 0 | 0 | 0 | 0.044 | 0.016 | 0.048 | 0.012 |
| Zoo | Avg | **0.002** | **0.002** | **0.002** | 0.061 | 0.018 | 0.069 | 0.005 |
| | StDev | 0 | 0 | 0 | 0.058 | 0.026 | 0.045 | 0.001 |

The superior performance of QBDA-DDO can be attributed to its specific algorithmic strategies. By integrating the strengths of the QBDA algorithm and incorporating the direction bias operator and adaptive step size mechanism, QBDA-DDO provides efficient and accurate search capabilities. These strategies guide individuals in more promising directions, leading to improved classification accuracy. Furthermore, both LBDA-DDO and SBDA-DDO also demonstrate competitive performance on their respective datasets, achieving relatively good results.

Table 13 shows the average number of selected features for the proposed algorithm and four comparison algorithms on the 14 datasets. LBDA-DDO, QBDA-DDO, and BASO achieved similar performance, obtaining the best results on four datasets. On the other hand, the BBA algorithm demonstrated its advantage in feature selection by obtaining the best results on two datasets. The comparable performance of LBDA-DDO, QBDA-DDO, and BASO indicates their effectiveness in selecting a reasonable number of features on different datasets.

Overall, the results in Table 13 emphasize the importance of algorithm selection in the feature selection task. This comparison provides us with the advantages and disadvantages of different algorithms and helps researchers and practitioners to choose the most suitable method according to the specific dataset and needs.

Table 14 presents a comprehensive comparison of our proposed BDA-DDO algorithm and four comparison algorithms (BGWO, BBA, BASO, and BGSA) based on their average fitness values across 14 datasets. The results clearly demonstrate that QBDA-DDO outperforms the other algorithms, achieving the best fitness values on all 13 datasets. Additionally, LBDA-DDO and SBDA-DDO also show strong competitiveness by obtaining the best results on seven and three datasets, respectively. This highlights the effectiveness

of the proposed algorithm compared to the four representative feature selection methods, demonstrating its superior performance.

Based on the analysis of the classification accuracy, number of selected features, and fitness, the proposed BDA-DDO algorithm consistently outperforms other algorithms across the majority of datasets. Among them, the QBDA-DDO algorithm demonstrates the highest performance, followed by LBDA-DDO and SBDA-DDO. The results highlight the effectiveness of our proposed method in selecting a smaller subset of features while achieving superior classification accuracy. Moreover, the proposed algorithm exhibits faster convergence speed and higher solution quality compared to three improved BDA algorithms. Furthermore, it proves to be highly competitive when compared to other feature selection algorithms.

## 5. Conclusions and Future Work

The purpose of this paper is to enhance the performance of the BDA algorithm through a hybrid approach. This goal has been successfully achieved by introducing three improvement mechanisms. Firstly, a novel differential operator, called the directed differential operator, is designed. Combining BDA with the directed differential operator provides a correct direction for the search process, resulting in faster convergence. Secondly, an adaptive update method is devised to enhance population diversity by updating the directed differential vector. Lastly, an adaptive step-updating mechanism is proposed to enhance the algorithm's exploration capability by adjusting the dragonfly step.

The proposed algorithm is evaluated on 14 mainstream datasets from the UCI library and compared with seven representative feature selection algorithms. The experimental results show that the proposed BDA-DDO algorithm outperforms LBDA on 10 datasets in terms of classification accuracy, while achieving the same accuracy on 4 datasets. Additionally, BDA-DDO selects smaller feature subsets than LBDA on all 14 datasets. Compared to QBDA, BDA-DDO achieves higher classification accuracy on 10 datasets and the same accuracy on some low-dimensional datasets (reaching 1). Moreover, BDA-DDO selects smaller feature subsets than QBDA on all 14 datasets. When compared with SBDA, BDA-DDO achieves higher classification accuracy on nine datasets and selects fewer features on 13 datasets. In conclusion, BDA-DDO demonstrates its superiority over the three BDA algorithms (LBDA, QBDA, and SBDA) by consistently achieving higher classification accuracy while selecting smaller feature subsets on most datasets. Moreover, when compared to four other typical feature selection algorithms (BGWO, BBA, BASO, and BGSA), it also achieves higher classification accuracy.

Although the proposed improvement mechanisms have successfully enhanced the performance of the BDA algorithm, there are still some limitations. Specifically, the algorithm's performance may be constrained when dealing with complex optimization scenarios, such as high-dimensional or large-scale datasets. Additionally, the feature selection problem is a multimodal problem, where the same number of features may correspond to different feature subsets. Currently, the algorithm may not be able to find all possible feature subsets. Moreover, the algorithm has not been tested in real-world applications, such as remote sensing tasks. Therefore, further research and experimentation is needed to address these issues and ensure the algorithm's applicability and effectiveness in practical scenarios.

In future research, our main focus will be to explore the practical applications of the proposed algorithm, particularly in the field of remote sensing [51]. Remote sensing datasets often exhibit high dimensionality and large scale, with a substantial number of features, samples, and classes, and may contain feature correlations or redundancies. These characteristics make the feature selection problem in remote sensing datasets more complex and challenging, requiring consideration of feature interactions and impacts, as well as the efficiency and stability of feature selection algorithms.

**Author Contributions:** Conceptualization, B.G.; Methodology, B.G. and Y.C.; Software, B.G.; Validation, B.G.; Investigation, B.G.; Writing—original draft, B.G.; Writing—review & editing, Y.C., T.L., H.L., Y.W., D.Z. and X.L.; Supervision, Y.C., T.L., H.L., Y.W., D.Z. and X.L.; Project administration, Y.C.; Funding acquisition, Y.C., T.L. and X.L. All authors have read and agreed to the published version of the manuscript.

**Funding:** This work is supported by the National Key Research and Development Program of China under Grant NO. 2020YFB1313900, the Scientific Research Foundation of Wuhan Institute of Technology under Grant NO. 21QD53, the Innovation Fund of Hubei Key Laboratory of Intelligent Robot under Grant NO. HBIRL202107 and NO. HBIR 202105, the Science and Technology Research Project of Education Department of Hubei Province under Grant NO. Q20221501, the Graduate Innovative Fund of Wuhan Institute of Technology under Grant NO. CX2022342, Shenzhen Science and Technology Program under Grant NO. JCYJ20200109115201707 and JCYJ20220818101408019.

**Data Availability Statement:** The datasets used in this study are available at the UCI database at http://archive.ics.uci.edu/datasets (accessed on 4 May 2022).

**Conflicts of Interest:** The authors declare no conflict of interest.

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
