# Peer review of "A Hybrid Binary Dragonfly Algorithm with an Adaptive Directed Differential Operator for Feature Selection"

_remotesensing, doi:10.3390/rs15163980_

Round 1
Reviewer 1 Report
This paper proposed a hybrid binary dragonfly algorithm with adaptive mechanism and differential evolution operator. The performance of the proposed algorithm was evaluated on 14 mainstream public datasets compared with BDA, BGWO, BBA, and BASO. There are still some points that need to be improved or discussed.
1. The algorithms applied to feature selection cited in the introduction and related work are not novel enough overall. Among the cited references, the number of papers published in the past three years is insufficient.
2. In BDA-DDO, several operators are designed to enhance the exploration capability in the late stage. But this improvement is not significant in Figure 1, 2, and 3.
3. The experiments are all performed on public datasets. It is recommended to apply this algorithm in practical domains with specialized datasets, especially in the field of remote sensing. Since the authors have described this application in the future research, please analyze the differences between the feature selection datasets for remote sensing and the 14 test datasets. Also, which datasets in the 14 test datasets are more similar to the datasets in the field of remote sensing?
4. The italics and subscripts of some variables need to be corrected. Including:
N in line 189, 301, j in line 192, s, a, c, f, e in line 206-207, 213-216, F in line 256-258, 284, D in line 265, T1 in line 268-271, R in line 300, k in line 329, and the variables in Algorithm 1 and Table 2.
Please check all variables in the manuscript carefully.
Minor editing of English language required.
Reviewer 2 Report
1. Literature review of this paper needs to be improved focusing on main work done in this paper.
2. Contribution and novelty of this work should be further justified
3. Simulation environment should be provided in details
4. Other typical methods should be compared to the proposed one.
5. The conclusion section needs to add the limitations of the proposed method
Reviewer want authors to fix grammatical issues in the article
Author Response
请参阅附件。

Reviewer 3 Report
This paper presents an improved dragonfly algorithm combined with a directed differential operator, named BDA-DDO.
Here are some suggestions on how to improve the paper:
1- The innovation and new contribution seem relatively limited and must be better underlined.
2- Numerical values of the superiority of the proposed method are shown in the conclusion sections.
3- A report on the background of the research is presented and no analysis is done on these researches and their results.
4- It is necessary first to determine the research motivations in relation to the current research conditions in this field. Then, the justification of the proposed method in this application should be presented.
5- The content of the paper is not standard; this factor makes the complexity of understanding.
6- The structure of the journal has not been appropriately followed.
7- The paper should be rewritten by a native English speaker.
The paper should be rewritten by a native English speaker.
Reviewer 4 Report
The manuscript titled "Hybrid Binary Dragonfly Algorithm with Adaptive Directed Differential Operator for Feature Selection" proposed a better version of DA, called BDA-DDO. BDA-DDO explores the data smarter and converges faster. The authors tested BDA-DDO on various datasets and found it to work better than other methods, including the original DA. The overall presentation and quality of the manuscript fit the scope of the special issue. The literature review was quite comprehensive as well. However, there are many language errors throughout the manuscript. English editing service is required for the manuscript.
There are too many language errors that need to be taken care of. For a few examples,
Line 3: to reducing;
Line 5: try to explore in the feature selection -> that try to explore the feature selection
Line 27: solve -> solving
Line 28: a few number of features from large numbers
Line 37: contributes
Line 45: becomes
Round 2
Reviewer 3 Report
Accept